

# qtQDA: quantile transformed quadratic discriminant analysis for high-dimensional RNA-seq data

Necla Koçhan[1], G. Yazgi Tutuncu[1], Gordon K. Smyth[2,3],
Luke C. Gandolfo[2,3] and Göknur Giner[2,4]

[1] Department of Mathematics, Izmir University of Economics, Izmir, Turkey
[2] Bioinformatics Division, The Walter and Eliza Hall Institute of Medical Research, Melbourne, VIC, Australia
[3] School of Mathematics and Statistics, University of Melbourne, Melbourne, VIC, Australia
[4] Department of Medical Biology, University of Melbourne, Melbourne, VIC, Australia

## ABSTRACT

Classification on the basis of gene expression data derived from RNA-seq promises to become an important part of modern medicine. We propose a new classification method based on a model where the data is marginally negative binomial but dependent, thereby incorporating the dependence known to be present between measurements from different genes. The method, called qtQDA, works by first performing a quantile transformation (qt) then applying Gaussian quadratic discriminant analysis (QDA) using regularized covariance matrix estimates.
We show that qtQDA has excellent performance when applied to real data sets and has advantages over some existing approaches. An R package implementing the method is also available on https://github.com/goknurginer/qtQDA.

## INTRODUCTION

Classification on the basis of gene expression data has the potential to become an important part of modern medicine, for example, for disease diagnosis and personalization of treatment. For example, consider breast cancer. This is a heterogeneous disease consisting of several distinct types, with each type being characterized, not necessarily by its morphological or clinical characteristics, but by its molecular characteristics, thereby making it difficult to diagnose the particular type affecting a patient (*Perou et al., 2000*). Moreover, the most effective treatment for each of these types may differ, for example, breast cancers that are growing in response to HER2 (human epidermal growth factor receptor 2 protein) can be treated with the targeted therapy drug trastuzumab, while ER+ (oestrogen hormone receptor positive) cancers may respond to hormone therapy that blocks oestrogen, on the other hand, triple negative (hormone receptor negative and HER2 negative) cancers do not respond to targeted therapy nor hormone therapy but respond to chemotherapy. Thus, if a woman has breast cancer, it is important to classify what type of cancer she has; this knowledge allows her treatment to be personalized,

Corresponding authors
Luke C. Gandolfo,
gandolfo@wehi.edu.au
Göknur Giner, giner.g@wehi.edu.au

increasing her chances of survival. One promising idea for achieving such classifications is to measure the pattern of gene expression in a patient sample and use this pattern of expression as data to classify which cancer type the patient has.

There are many ways of measuring gene expression. One common approach, due to its numerous advantages, is RNA-sequencing (RNA-seq) which measures gene expression across the whole genome simultaneously (see *Mardis, 2008*; *Wang, Gerstein & Snyder, 2009*). RNA-seq involves three main steps: (1) mRNA is obtained from a sample and broken into millions of short segments; (2) these mRNA segments are converted into cDNA; and (3) these cDNA segments are sequenced using next-generation sequencing. The resulting sequence data is then mapped to genomic regions of interest, typically genes, and the number mapping to each region is counted. Thus, in essence, RNA-seq data consists of *counts*: for each gene we obtain a non-negative integer count which quantifies the gene's expression level; roughly speaking, the larger the count the higher the level of expression.

Several approaches have been proposed for classifying RNA-seq data. General machine learning approaches have been investigated, for example, support vector machines (SVMs) and k-Nearest Neighbour (kNN) classifiers, and general regression approaches have also been applied, for example, logistic regression (see *Tan, Petersen & Witten, 2014*; *Zararsiz et al., 2017b*). Others have focused on modeling the data more directly. For example, *Witten (2011)* proposed the PLDA method, which models the counts using the Poisson distribution, while *Dong et al. (2016)* proposed the NBLDA method, which instead models the counts using the negative binomial distribution, thereby taking into account the overdispersion known to be present in RNA-seq data on biological replicates. Others still have proposed transforming the counts, for example, using a log transformation, so that variations on traditional classification techniques become available, for example, Gaussian classification. The best example of this sort is the method voomDLDA (*Zararsiz et al., 2017a*). One common feature of these direct modeling approaches is that they are, in classification terminology, "naïve": they assume that measurements on the features used for classification, that is, the genes, are statistically *independent*.

However, this independence assumption is very unrealistic, since genes are typically involved in networks and pathways, implying that a particular gene's expression level is likely to be correlated with the expression level of other genes. Moreover, some have argued, for example, *Zhang (2017)*, that the assumption of independence has a non-ignorable impact on our ability to classify: it causes bias in estimated discriminant scores, making classification inaccurate. Given this, some have focused on models for the data which incorporate dependence between genes. For example, *Sun & Zhao (2015)* proposed the SQDA method which models log-transformed counts with the multivariate normal distribution using regularized estimates of covariance matrices, which are assumed to be different for each class. More recently, *Zhang (2017)* developed a Bayesian approach where the data is modeled using a (multivariate) Gaussian copula.

In this article we propose a new classification method for RNA-seq data based on a model where the counts are marginally negative binomial but dependent. Like previous

work, we use the multivariate normal distribution for classification, where each class is assumed to have its own covariance matrix. However, our approach has two key differences: (1) instead of modeling log-transformed counts, we model quantile transformed counts; and (2) we use a novel application of a powerful regularization technique for covariance matrix estimation. We call the method qtQDA: quantile transformed (qt) quadratic discriminant analysis (QDA). We demonstrate the performance of the method by applying it to several real data sets, showing that it performs better than, or on par with, existing methods. qtQDA has advantages over some existing approaches, and an R package `qtQDA` implementing the method is available on https://github.com/goknurginer/qtQDA.

# METHODOLOGY

## The model

First we describe the model underpinning qtQDA. Suppose we wish to classify data into one of $K$ distinct classes on the basis of $m$ genes (i.e., features). Let $\mathbf{X}^{(k)} = \left[X_1^{(k)}, X_2^{(k)}, \ldots, X_m^{(k)}\right]^T$ be a random vector from the $k$th class where $X_i^{(k)}$ denotes the count for gene $i$. Like others, for example, NBLDA and the method of *Zhang (2017)*, we assume the counts are marginally negative binomial, that is,

$$X_i^{(k)} \sim NB(\mu_i^{(k)}, \phi_i^{(k)}), \tag{1}$$

where $\mu_i^{(k)}$ and $\phi_i^{(k)}$ are the mean and dispersion for gene $i$, respectively (strictly speaking, $\mu_i^{(k)}$ depends on the "library size," but for the purposes of clarity, this complication is addressed later). Note that, for non-zero dispersion,

$$\mathrm{Var}(X_i^{(k)}) = \mu_i^{(k)} + \phi_i^{(k)}(\mu_i^{(k)})^2 > \mu_i^{(k)},$$

that is, the data is over-dispersed relative to Poisson variation, consistent with known properties of RNA-seq data on biological replicates (see *McCarthy, Chen & Smyth, 2012*). Unlike others, however, we suppose that $\mathbf{X}^{(k)}$ is generated by the following process:

1. Let $\mathbf{Z}^{(k)}$ be an $m$-vector from a multivariate normal distribution: $\mathbf{Z}^{(k)} \sim \mathrm{MVN}(\mathbf{0}, \mathbf{\Sigma}_k)$, where $Z_i^{(k)} \sim N(0,1)$.

2. Then let the $i$th component of $\mathbf{X}^{(k)}$ be the transformed random variable

$$X_i^{(k)} = F_k^{-1}\{\Phi(Z_i^{(k)})\}, \tag{2}$$

where $\Phi$ is the standard normal distribution function and $F_k$ is the $NB(\mu_i^{(k)}, \phi_i^{(k)})$ distribution function.

We make two observations. Firstly, observe that the transformation in Eq. (2) generates a vector $\mathbf{X}^{(k)}$ with the negative binomial margins specified in Eq. (1). This is a consequence of the following elementary fact from probability theory: if $F$ and $G$ are distribution functions, and $X$ has distribution function $F$, then the transformed variable $G^{-1}\{F(X)\}$ has distribution function $G$ (see *Lange, 2010*, p. 432). We call the kind of transformation invoked here a *quantile transformation*. Note that, given the discreteness of the

negative binomial distribution, the ambiguity of $F_k^{-1}$ is obviated by imposing that $F_k^{-1}(q) = \inf\{x : F_k(x) \geq q\}$, for $q \in [0, 1]$.

Secondly, observe that the negative binomial components of $\mathbf{X}^{(k)}$ are not independent: the underlying MVN distribution, with a dependence structure encoded in $\mathbf{\Sigma}_k$, generates a dependence structure between the components of $\mathbf{X}^{(k)}$. Note especially that each class is assigned a different covariance matrix. As *Sun & Zhao (2015)* have suggested, since the presence of disease, and different disease types, leads to "rewiring" of genetic networks, and hence changes in gene associations, assuming a different covariance matrix for each class is likely to lead to better classifications. Finally, note that while the model specified by the process above is reminiscent of the Gaussian copula model of *Zhang (2017)*, the two models are quite different.

## Classification

We now turn to how the model above is used for classification. Suppose we observe $\mathbf{x}^* = \left[x_1^*, x_2^*, \ldots, x_m^*\right]^T$ from unknown class $y^*$, where $y^* \in \{1, 2, \ldots, K\}$. For each class we apply the inverse of the quantile transformation Eq. (2) to the components of $\mathbf{x}^*$ to produce a new vector $\mathbf{z}^{*(k)}$, that is, where

$$z_i^{*(k)} = \Phi^{-1}\{H_k(x_i^*)\} \tag{3}$$

and $H_k$ is a continuity-corrected version of $F_k$. Here $H_k$ is defined by

$$H_k(x_i^*) = \Pr(X < x_i^*) + 0.5 \times \Pr(X = x_i^*),$$

where $X$ is a NB $(\mu_i^{(k)}, \phi_i^{(k)})$ distributed random variable, and $H_k(X)$ is more nearly uniformly distributed than $F_k(X)$ itself (*Routledge, 1994*). The transformation from $x_i^*$ to $z_i^{*(k)}$ is implemented by the `zscoreNBinom` function in the R package `edgeR` (see below).

Once this transformation has been made, given the assumptions of the model, traditional QDA now becomes available, as follows. Under the model, if $\mathbf{x}^*$ is from the $k$th class then $\mathbf{z}^{*(k)}$ is an observation from the MVN$(\mathbf{0}, \mathbf{\Sigma}_k)$ distribution. Thus, by Bayes theorem, the posterior probability that $\mathbf{x}^*$ belongs to the $k$th class is

$$\Pr(y^* = k|\mathbf{x}^*) \propto f_k\left(\mathbf{z}^{*(k)}\right)\pi_k, \tag{4}$$

where $\pi_k$ is the prior probability that $\Pr(y^* = k)$, and $f_k$ is the density

$$f_k(\mathbf{v}) = \frac{1}{(2\pi)^{m/2}|\mathbf{\Sigma}_k|^{1/2}} \exp\left\{-\frac{1}{2}\mathbf{v}^T\mathbf{\Sigma}_k^{-1}\mathbf{v}\right\}$$

evaluated at $\mathbf{z}^{*(k)}$. We classify $\mathbf{x}^*$ into the class that maximizes this posterior probability. It is worth noting that since maximizing Eq. (4) is equivalent to maximizing $\log \Pr(y^* = k|\mathbf{x}^*)$, this classification rule entails the following (quadratic) discriminant function:

$$\delta_k(\mathbf{x}^*) = -\frac{1}{2}\mathbf{u}_k^T\mathbf{u}_k + \log \pi_k,$$

where $\mathbf{u}_k = \mathbf{\Sigma}_k^{-1/2}\mathbf{z}^{*(k)}$, which has the following insightful interpretation: for a given class $k$, the further the vector $\mathbf{u}_k$ is from the origin, the less likely $\mathbf{x}^*$ is to belong to that class.

## Parameter estimation

To use the classifier in practice the parameters of the underlying model need to be estimated, that is, the classifier needs to be "trained." Specifically, for each gene $i = 1, 2, \ldots, m$ and class $k = 1, 2, \ldots, K$, we need to estimate the negative binomial means $\mu_i^{(k)}$ and dispersions $\phi_i^{(k)}$, to parametrize the quantile transformation Eq. (3), and we need to estimate the covariance matrix $\Sigma_k$ of the transformed variables, so QDA can be performed with Eq. (4). For each class $k$, suppose we have a set of $n$ RNA-seq samples $\mathbf{x}_1^{(k)}, \mathbf{x}_2^{(k)}, \ldots, \mathbf{x}_n^{(k)}$ known to belong to class $k$.

To estimate the negative binomial parameters we use the methodology implemented in the R package edgeR (*McCarthy, Chen & Smyth, 2012*; *Chen, Lun & Smyth, 2014*) which is extremely fast and reliable, and offers three sophisticated approaches for dispersion estimation. Maximum likelihood estimates (MLEs) of the gene means are found by fitting a negative binomial generalized linear model (GLM) with logarithmic link function:

$$\log \mu_{ij}^{(k)} = \beta_i^{(k)} + \log N_j,$$

where $\log N_j$ is a model offset and $N_j$ is the "library size" for sample $j$, that is, the total counts $\Sigma_i x_i^{(k)}$ across all observed genes in the RNA-seq sample. The resulting gene mean estimates are then given by $\hat{\mu}_{ij}^{(k)} = N_j \exp(\hat{\beta}_i^{(k)})$. Note that the use of a GLM with $\log N_j$ as an offset allows us to avoid the use of "size factors" which are commonly employed in other Poisson or negative binomial based methods to scale counts to account for differences in library sizes (e.g., PLDA, NBLDA, and the method of *Zhang (2017)*).

The dispersion $\phi_i^{(k)}$ for each gene is estimated using the Cox-Reid adjusted profile likelihood (APL) function:

$$\mathrm{APL}_i(\phi_i^k) = \mathscr{S}(\phi_i^k) - \frac{1}{2} \log \det(\mathscr{I}_i^{(k)}), \tag{5}$$

where $\ell$ is the log-likelihood and $\mathscr{I}_i^{(k)}$ is the Fisher information of $\beta_i^{(k)}$, both functions being evaluated at the MLE $\hat{\beta}_i^{(k)}$. This modified likelihood function adjusts for the fact that the gene mean is estimated from the same data, thereby reducing the bias of the MLE of $\phi_i^{(k)}$. Instead of simply maximizing Eq. (5), however, to achieve even better dispersion estimates, an approximate empirical Bayes strategy is applied, where the APL for each gene is substituted by a weighted sum of APLs from carefully chosen sets of genes, resulting in "information sharing" between genes, and thereby better dispersion estimates for individual genes (see *Chen, Lun & Smyth, 2014* for details). Using different variations of this general approach, edgeR offers three kinds of dispersion estimates: "common," "trended," and "tag-wise." By default, qtQDA uses the "tag-wise" dispersion estimates (but, the user is free to choose any of these kinds).

Once the negative binomial parameters have been estimated we apply the quantile transformation Eq. (3) to the components of the RNA-seq sample vectors to produce a corresponding set of transformed vectors $\mathbf{z}_1^{(k)}, \mathbf{z}_2^{(k)}, \ldots, \mathbf{z}_n^{(k)}$ where, under the assumed

model, $\mathbf{z}_j^{(k)} \sim \text{MVN}(\mathbf{0}, \Sigma_k)$. To estimate the covariance matrix $\Sigma_k$, we begin by calculating the standard estimate:

$$\hat{\Sigma}_k = \frac{1}{n-1} \sum_{j=1}^{n} \left\{ \mathbf{z}_j^{(k)} - \bar{\mathbf{z}}^{(k)} \right\} \left\{ \mathbf{z}_j^{(k)} - \bar{\mathbf{z}}^{(k)} \right\}^T,$$

where $\bar{\mathbf{z}}^{(k)} = \sum_{j=1}^{n} \mathbf{z}_j^{(k)}/n$. As it stands, however, this estimate is not useful in the present context where the data is typically "high dimensional," that is, where the number of genes used for classification will be approximately the same or greater than the number of samples (i.e., $m \approx n$ or $m > n$). In such situations this standard covariance matrix estimate is known to perform poorly (see *Tong, Wang & Wang, 2014*). To remedy this, we regularize the standard estimate using the approach developed in *Schäfer & Strimmer (2005)* and *Opgen-Rhein & Strimmer (2007)* which is implemented in the R package `corpcor` (see also *Strimmer, 2008*). The `corpcor` method separately shrinks the corresponding correlation estimates $\hat{\rho}_{ii'}$ toward zero and the variance estimates $\hat{v}_i$ toward their median to produce the regularized estimates

$$\tilde{\rho}_{ii'} = (1 - \lambda_1)\hat{\rho}_{ii'}$$

$$\tilde{v}_i = \lambda_2 v_{\text{median}} + (1 - \lambda_2)\hat{v}_i$$

where the shrinkage intensities are estimated via

$$\hat{\lambda}_1 = \frac{\sum_{i \neq i'} \widehat{\text{Var}}(\hat{\rho}_{ii'})}{\sum_{i \neq i'} \hat{\rho}_{ii'}^2} \quad \text{and} \quad \hat{\lambda}_2 = \frac{\sum_{i=1}^{m} \widehat{\text{Var}}(v_i)}{\sum_{i=1}^{m} (v_i - v_{\text{median}})^2}.$$

The regularized covariance matrix estimate $\tilde{\Sigma}_k$ then has entries $[\tilde{\Sigma}_k]_{ii'} = \tilde{\rho}_{ii'}\sqrt{\tilde{v}_i \tilde{v}_{i'}}$. This estimate has two excellent statistical properties: (1) it is always positive definite and well conditioned (making the inverse computable); and (2) it is guaranteed to have minimum mean squared error, which is a consequence of an important result proved by *Ledoit & Wolf (2003)*. Moreover, since the shrinkage intensities are calculated with analytic formulas, the estimate also has two significant practical advantages: (1) it is computationally very fast to compute; and (2) it does not require any "tuning" parameters. We note that the `corpcor` approach to covariance matrix regularization is quite different to the computationally intensive approach used in SQDA (*Sun & Zhao, 2015*). Note also that, while the `corpcor` method has previously been used for classification of gene expression data from microarrays (see *Xu, Brock & Parrish, 2009*), we appear to be the first to use it for RNA-seq data.

The final parameter needed for classification with Bayes theorem Eq. (4) is the prior probability of belonging to the $k$th class $\pi_k = \text{Pr}(y^* = k)$. This probability can either be specified by the user, for example, if epidemiological knowledge is available, or estimated directly from the training data using

$$\hat{\pi}_k = \frac{\sum_{j=1}^{n'} I\{y_j = k\}}{n'},$$

where $I\{\cdot\}$ is the indicator function, and $n'$ is the total number of samples in all $K$ classes.

## Feature selection

Lastly, we turn to the question of which genes to use for classification. When RNA-seq is performed we typically obtain data on more than 20,000 genes. A vast number of these genes, however, will not be informative for the purposes to distinguishing between different classes. We therefore employ the following simple strategy for selecting $m$ genes for classification: (1) we filter genes with low expression across all samples; (2) for each remaining gene we perform a likelihood ratio test (LRT) to test for genes differentially expressed between groups; (3) a list of genes is made, sorted by LRT statistic; (4) finally, the top $m$ genes from this list is used for classification. As with negative binomial parameter estimation, this strategy is implemented using edgeR. Others have adopted essentially the same gene selection strategy, for example, NBLDA, SQDA, and the method of *Zhang (2017)*.

## RESULTS

To assess the performance of qtQDA we apply it to three publicly available data sets:

1. *Cervical cancer data* (see *Witten et al., 2010*). This consists of two classes, cancer and non-cancer, each with 29 samples. Each sample consists of counts for 714 different microRNAs obtained using RNA-seq.

2. *Prostate cancer data* (see *Kannan et al., 2011*). This consists of two classes, 20 samples from cancer patients and 10 samples from benign matched controls. Each sample consists of RNA-seq data for the whole transcriptome.

3. *HapMap data* (see *Montgomery et al., 2010*; *Pickrell et al., 2010*). The data considered here consists of two of the HapMap populations: CEU (Utah residents with Northern and Western European ancestry) and YRI (Yoruba in Ibadan, Nigeria). There are 60 CEU samples and 69 YRI samples, each consisting of RNA-seq data for the whole transcriptome, and all from "healthy" individuals.

These data sets are very common in the RNA-seq classification literature (see *Witten, 2011*; *Tan, Petersen & Witten, 2014*; *Dong et al., 2016*; *Zhang, 2017*). Using these data sets, we also compare the performance of qtQDA to a number of general machine learning classifiers and specialized RNA-seq classifiers (corresponding R packages used for our analysis are listed in brackets):

- SVM (e1071)
- kNN (e1071)
- Logistic regression (glmnet)
- PLDA (PoiClaClu)
- NBLDA (http://www.comp.hkbu.edu.hk/xwan/NBLDA.R)
- voomDLDA (MLSeq)
- SQDA (SQDA)

For logistic regression, we use the GLMnet method proposed in *Friedman, Hastie & Tibshirani (2010)* since this is one of the best representatives of this approach. This method
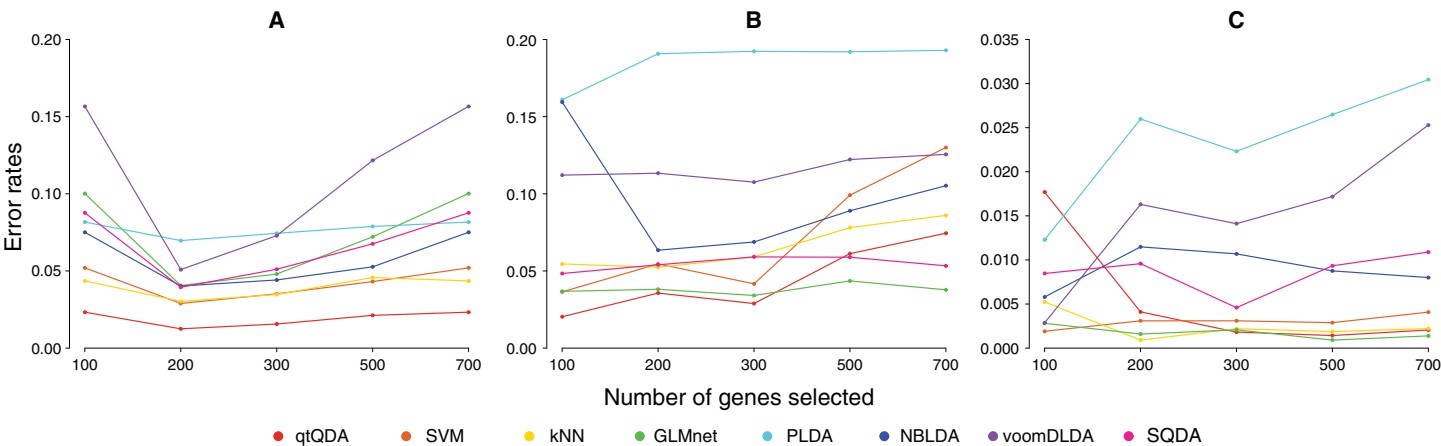

**Figure 1 Error rate vs genes selected.** These plots show classification error rate as a function of the number of genes chosen for classification for the (A) cervical cancer, (B) prostate cancer, and (C) HapMap data sets.

uses a "lasso" (i.e., $\ell_1$) penalty in the log-likelihood function which thus overcomes many of the problems with logistic regression in high-dimensional settings (see *Tan, Petersen & Witten, 2014*) and encourages regularized regression coefficients, that is, shrunken to zero. For the SVM method we used a radial basis kernel, and for the kNN method we used $k = 1$, 3, and 5 (but only report results for $k = 1$ since this consistently performed best), and both methods were applied to log transformed counts. We apply all methods as recommended in their documentation and any "tuning" parameters were chosen with the cross-validation tools provided in the corresponding software package or chosen with our own cross-validation. The Gaussian copula method of *Zhang (2017)* has no publicly available implementation.

For evaluation, we estimated the true error rate, that is, the rate at which false classifications are made, using the following bootstrap procedure: (1) each data set is randomly divided into two parts, one part consisting of 70% of the data, put aside for training the classifier, and one part consisting of 30% of the data, used as a test set to apply the trained classifier from which an error rate is recorded; (2) this is repeated 1,000 times and the error rates from each iteration is averaged to produce an estimate of the true error rate. This is the same procedure used by *Dong et al. (2016)* and *Zhang (2017)*. We estimated the error rates for $m = 100$, 200, 300, 500, 700 genes, where these genes are selected using the procedure detailed in the previous section.

Results are shown in Fig. 1 and Table 1. We see that qtQDA performs best for both cancer data sets, achieving the lowest error rate at 200 genes for the cervical cancer data and 100 genes for the prostate cancer data. Interestingly, for the cervical cancer data, qtQDA uniformly achieves the smallest error rate. For the HapMap data, qtQDA essentially performs as well as the SVM, kNN, and logistic regression classifiers. We note that even though these classifiers have similar performance, we think qtQDA or logistic regression would be preferred, at least in a medical context, since these classifiers do more than merely assign a sample to a particular class: they also provide a posterior probability of belonging to each class. This is important in a medical context where the

**Table 1 Minimum error rates.** This table shows the minimum error rates achieved for each classifier in each data set. The number of genes used to obtain this minimum error rate is reported in brackets.

| Method | Cervical cancer | Prostate cancer | HapMap |
| --- | --- | --- | --- |
| qtQDA | 0.0125 (200) | 0.0203 (100) | 0.0018 (300) |
| SVM | 0.0276 (100) | 0.0364 (100) | 0.0014 (500) |
| kNN | 0.0277 (100) | 0.0523 (200) | 0.0009 (200) |
| GLMnet | 0.0406 (200) | 0.0341 (300) | 0.0009 (500) |
| PLDA | 0.0608 (100) | 0.1609 (100) | 0.0123 (100) |
| NBLDA | 0.0402 (200) | 0.0634 (200) | 0.0058 (100) |
| voomDLDA | 0.0425 (100) | 0.1076 (300) | 0.0029 (100) |
| SQDA | 0.0318 (100) | 0.0483 (100) | 0.0046 (300) |

different treatments or further diagnostic procedures which could be prescribed, following a classification, may be associated with very different risks.

## DISCUSSION

Early investigations into classification with gene expression data from microarrays, for example, *Dudoit, Fridlyand & Speed (2002)*, showed that making the (unrealistic) assumption of independence between measurements from different genes can still lead to classifiers with good performance. Our results, however, seem to suggest that incorporating dependence between genes can lead to even better performance, at least for RNA-seq data.

Our method has two key advantages. Firstly, unlike some approaches (e.g., kNN, GLMnet, PLDA, SQDA), qtQDA does not have any "tuning" parameters which need to be chosen with cross-validation, thus making it more straightforward to apply in practice. Secondly, in comparison to approaches which take gene dependence into account, for example, SQDA and the method of *Zhang (2017)*, qtQDA is computationally much faster. SQDA adopts a computationally intensive method for covariance matrix regularization. In an effort to reduce the required computation, the authors impose a block diagonal structure on the covariance matrix where each block is assumed to be the same size (but which needs to be determined by cross-validation), simplifications which even the authors acknowledge are unrealistic (e.g., under these assumptions the order of the genes used for classification matters). Yet, despite these simplifications, extensive computation is still required, making the method very slow. On the other hand, the regularization approach applied in qtQDA requires no special assumptions for the covariance matrix and requires minimal computation since the regularized estimate is obtained with analytic formulas. The Gaussian copula method of *Zhang (2017)* is also computationally intensive, but for a different reason: it is cast in a Bayesian framework and requires a Metropolis-Hasting algorithm, in combination with Gibbs sampling, for parameter estimation. As the author acknowledges, the computations required are time consuming even when implemented in a fast language like C++.

As *Dudoit, Fridlyand & Speed (2002)* points out, there are three related statistical problems in the area of classifying disease with gene expression data: (1) identifying new disease subclasses, that is, cluster analysis; (2) classifying samples into known disease

classes, that is, discriminant analysis; and (3) identifying "marker" genes that characterize different disease subclasses, that is, variable selection. This paper has firmly focused on problem (2), which is why it was sufficient to evaluate classifier performance solely in terms of error rate and not sparsity, that is, the number of features used to make classifications. The feature selection method we proposed, while likely to deliver many genes informative for classification, is clearly too simplistic to deliver *only* those genes which are informative for distinguishing between classes. Thus, future research will aim at developing a sparse version of qtQDA, involving some level of regularization for features, that is, identifying less informative features and reducing their influence to zero (e.g., like the GLMnet logistic regression classifier). A sparse qtQDA may also help address problem (3) above, the answer to which has practical advantages, for example, knowing which subset of genes need to be measured for effective classification, and theoretical advantages, for example, obtaining insight into the underlying biological process driving the disease (or subclass) in question. A sparse qtQDA may also deliver a further bonus: it may lead to a better answer to problem (2), that is, to even better disease classifications.

## CONCLUSION

We have proposed a new classification method for RNA-seq data based on a model where the data is marginally negative binomial but dependent, thereby incorporating dependence between genes. The method works by first performing a quantile transformation then applying Gaussian QDA, where each class is assumed to have its own covariance matrix. The classifier is trained by using the sophisticated `edgeR` methodology for negative binomial parameter estimation, to parametrize the quantile transformation, and by using the powerful `corpcor` methodology for regularized covariance matrix estimation, so that effective QDA can be performed on the transformed data. We have shown that, when applied to real data sets, the classifier has excellent performance in comparison to other methods, and has two key advantages which makes it easy to apply in practice: (1) it does not have any tuning parameters; and (2) it is computationally very fast. An R package called `qtQDA` implementing the method is also available on https://github.com/goknurginer/qtQDA.

## ACKNOWLEDGEMENTS

We thank Prof. Terry Speed for helping us clarify the differences between our qtQDA model and the Gaussian copula model of *Zhang (2017)*, for recommending the `corpcor` covariance matrix regularization method, and for commenting on a draft manuscript.

### Funding

This work was supported by the Scientific and Technical Research Council of Turkey (TUBITAK 2214/A—1059B141601270) and by the Australian National Health and Medical Research Council (Program Grant 1054618 and Fellowship 1154970 to Gordon K.

Smyth), the Cancer Therapeutics CRC, Victorian State Government Operational Infrastructure Support and Australian Government NHMRC IRIIS. Funding for the article processing fee was provided by Smyth Lab funds. The funders had no role in study design, data collection and analysis, decision to publish, or preparation of the manuscript.

## Grant Disclosures

The following grant information was disclosed by the authors:
The Scientific and Technical Research Council of Turkey (TUBITAK): 2214/A—1059B141601270.
The Australian National Health and Medical Research Council (Grant and Fellowship): 1054618 and 1154970.
Cancer Therapeutics CRC, Victorian State Government Operational Infrastructure Support and Australian Government NHMRC IRIIS.
Smyth Lab funds.

## Competing Interests

The authors declare that they have no competing interests.

## Author Contributions

- Necla Koçhan conceived and designed the experiments, performed the experiments, analyzed the data, contributed reagents/materials/analysis tools, prepared figures and/or tables, authored or reviewed drafts of the paper, approved the final draft.
- G. Yazgi Tutuncu conceived and designed the experiments, authored or reviewed drafts of the paper, approved the final draft.
- Gordon K. Smyth conceived and designed the experiments, contributed reagents/materials/analysis tools, authored or reviewed drafts of the paper, approved the final draft.
- Luke C. Gandolfo conceived and designed the experiments, performed the experiments, analyzed the data, contributed reagents/materials/analysis tools, prepared figures and/or tables, authored or reviewed drafts of the paper, approved the final draft.
- Göknur Giner conceived and designed the experiments, performed the experiments, analyzed the data, contributed reagents/materials/analysis tools, prepared figures and/or tables, authored or reviewed drafts of the paper, approved the final draft.

## Data Availability

qtQDA R package is available on https://github.com/goknurginer/qtQDA.

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
