# Peer review of "qtQDA: quantile transformed quadratic discriminant analysis for high-dimensional RNA-seq data"

_PeerJ, doi:10.7717/peerj.8260_

## Round 0.1 · original submission · Minor Revisions

The manuscript has been assessed by two reviewers, and they agree on the fact that there are a few points that need to be addressed. We would be glad to consider a substantial revision of your work, where the reviewers' comments will be carefully addressed one by one.

Reviewer 1 ·

Basic reporting

The article is well written, with the standard sections.

A comprehensive review of current methods is done, with a good comparison among them.

In several places it is said that a R library with the implementation is available, but the name is not given, although it is straightforward (qtQDA).

And the web page is also not given : https://github.com/goknurginer/qtQDA

Experimental design

The research question is well defined, it is important to have a way to classify samples based on its rnaseq results. An in-deph mathematical study is done.

But, in my opinion, it would be worthwhile to add an algorithm for the case where there are paired samples, that is, tumoral and healthy samples from the same person. That would make the results more precise, but would need a different processing.

And it would be interesting to know how it would be used if the whole RNA is used, so the repetitive elements are incorporated to the data used. But I can understand that that would be material for a new paper.

Validity of the findings

The cases and the comparison with previous work is very well done, and the results supports the authors in their findings.
The advantages are clearly exposed, but I think that being computationally fast is not a big advantage, as the whole processing needed to reach the stage where the classification is done is much bigger, as the reads must be aligned, etc, and this takes a lot of computing power.

But I would like to see an explanation of why so many genes are used for classification. Why not use only a few genes ? that would make the use of PCR technology feasible, as less genes would need to be used in the classification, making it more economical.

Additional comments

None

Reviewer 2 ·

Basic reporting

In this manuscript the authors described a new classification method, qtQDA, that is based on a model where the data is marginally negative binomial but dependent. The authors compared their method with other existing approaches and found that qtQDA performs better in most cases than the other methods.
In my opinion this is a good work and the manuscript is written in a clear and concise manner. I tested the R package with the implementation of the method, and everything worked well.

Experimental design

no comment

Validity of the findings

no comment

Additional comments

My only recommendation is that a user’s guide is included in the R package with the different options explained and practical examples given, similar to the edgeR users guide. I realize that each function is explained in the help section of the package but the integration of all the information in a single downloadable pdf file would greatly benefit the users. I especially would like to see a more detailed explanation on the three options for the dispersion estimates that even though are part of the edgeR package and are explained in their user’s guide, significantly impact the usage of the qtQDA method.

---

## Round 0.2 · accepted · Accept

Thank you very much for the submission of a revised version of your paper. I have gone through the track-changes manuscript and rebuttal letter and see that the authors addressed the reviewers' concerns and substantially improved the content of the manuscript. So, based on my own assessment as an academic editor, no further revisions are required, and the manuscript may be now accepted for publication in its current form.

Reviewer 1 ·

Basic reporting

The comments have been addressed in the text.

Experimental design

No changes from the previous review, it can be published in its current form.

Validity of the findings

I continue thinking that selecting automatically a smaller set of genes would provide an interesting view of the difference among the samples, providing the opportunity to know better the underlying mechanism causing that difference.
But I understand that this can be out of the scope of the present work.

Additional comments

None.

Reviewer 2 ·

Basic reporting

no comment

Experimental design

no comment

Validity of the findings

no comment

Additional comments

The authors have answered my comments and I have nothing else to add.